# Co-Culturing Seaweed with Scallops Can Inhibit the Occurrence of *Vibrio* by Increasing Dissolved Oxygen and pH

**DOI:** 10.3390/plants14030334

**Published:** 2025-01-23

**Authors:** Shuangshuang Zhang, Wei Lin, Sijie Liang, Guangda Sun, Jianting Yao, Delin Duan

**Affiliations:** 1Shandong Province Key Lab of Experimental Marine Biology, Institute of Oceanology, Chinese Academy of Sciences, Qingdao 266071, China; s17664080568@163.com (S.Z.); wlin@qdio.ac.cn (W.L.); lsjne114@163.com (S.L.); 2Lab for Marine Biology and Biotechnology, Qingdao Marine Science and Technology Center, Qingdao 266237, China; 3University of Chinese Academy of Sciences, Beijing 100049, China; 4Changdao Dongxing Aquaculture Co., Ltd., Changdao 265800, China; cddx3218314@126.com

**Keywords:** seaweed, scallop, co-culture system, *Vibrio* inhibition, microbial diversity

## Abstract

Seaweeds are critically important for the maintenance of biodiversity in marine aquaculture ecosystems, as they can inhibit the growth of *Vibrio*. Here, we determined the optimal environmental parameters for co-culturing green macroalgae (*Ulva pertusa*) and red macroalgae (*Gracilariopsis lemaneiformis*) with Chinese scallop (*Chlamys farreri*) by measuring dissolved oxygen (DO), pH, and the strength of *Vibrio* inhibition under laboratory conditions and validating the effectiveness of this optimal co-culture system from the perspectives of nutrient levels, enzyme activities, and microbial diversity. The results show that co-culturing 30 g of seaweed and three scallops in 6 L of seawater with aeration in the dark (1.25 L min^−1^, 12:12 h L:D) significantly decreased the number and abundance of *Vibrio* after 3 days. The activities of superoxide dismutase, catalase, pyruvate kinase, and lactate dehydrogenase in *C. farreri* were significantly higher, indicating that its immune defense and metabolism enhanced in this optimal co-culture system. High DO and pH levels significantly decreased the alpha diversity of microorganisms, and the abundance of pathogenic microorganisms decreased. The optimal co-culture system was effective for the control of vibriosis. Generally, our findings suggest that seaweeds could be used to enhance the aquaculture environment by conferring healthy and sustainable functions in the future.

## 1. Introduction

Seaweeds play key roles in the maintenance of the biodiversity and ecological functions of marine ecosystems by improving the nutrient-rich seawater system [1,2,3]. *Ulva pertusa* can efficiently absorb nutrients in eutrophic marine environments and can be used to ameliorate eutrophication [4,5]. Many studies have characterized the antimicrobial properties of seaweeds, and live *Ulva fasciata* tissue can inhibit 99% of *Vibrio parahaemolyticus* [6]. Fucosterol and hexadec-4-enoic acid from *Sargassum longifolium* inhibit the growth of *V. parahaemolyticus*, *Vibrio vulnificus*, and *Vibrio harveyii* [7]. Pang et al. [8] found that the rapid reduction in culturable *V. parahaemolyticus* in the culture of live *Grateloupia turuturu* under light is caused by hydrogen peroxide or other reactive oxygen species (ROS) produced by algae. Lin et al. [9] found that the affinity of commensal bacteria for marine microalgae was stronger than that of *Vibrio*; commensal bacteria preferentially occupy the marine ecosystem, leading to *Vibrio* inhibition. The composition of microorganisms in the algal micro-environment is complex and diverse, forming a unique ecological relationship with the algae. The microorganisms in the algal micro-environment can produce some unique metabolites (such as antibiotics, toxins, etc.) and release them into the algal micro-environment to enhance the disease resistance of the algae itself [10]. Chu et al. [11] isolated seven strains of active bacteria with an inhibitory effect on *Vibrio alginolyticus* from the surface of kelp. *Vibrio* abundance was negatively correlated with dissolved oxygen (DO), which indicated that increases in DO can inhibit *Vibrio* growth [12,13,14].

In 2020, global scallop production reached 1746.4 thousand tonnes and annual scallop production was about 7.26 thousand tonnes, which comprised 9.8% of mollusk aquaculture production [15]. In China, the three primary scallop species cultivated in aquaculture systems are the Chinese scallop (*Chlamys farreri*), the bay scallop (*Argopecten irradians*), and the Japanese scallop (*Mizuhopecten yessoensis*) [16]. Among these, *Chlamys farreri* is a commercially important scallop for aquaculture in China [17]. Currently, raft culture, string-ear suspension culture, and bottom seeding culture are the main methods for breeding *C. farreri* [18]. In addition, there are also studies on the comprehensive culture of shellfish to prevent and deal with the key problem of large-scale mortality [19], which is a representative model of green development of the aquaculture industry. By introducing filter feeding shellfish and macroalgae, and using the filter feeding of shellfish and the photosynthetic absorption of nutrients by macroalgae, the environmental pressure caused by cage culture can be reduced and the load of organic matter and nutrients can be reduced [20]. The production of scallops in China has shown a declining tendency over the last few years, which was 2.01 million tonnes in 2017, 1.92 million tonnes in 2018, and 1.83 million tonnes in 2019 [21]. High temperatures, the destruction of the microbial community, and the proliferation of *Vibrio* can induce the mass death of Chinese scallop (*C. farreri*) [22,23]. *Vibrio* is widespread in marine aquaculture systems and has a major effect on the survival of scallops [24,25], as it can produce extracellular toxins and enzymes that are lethal to scallops [26,27]. The vibriosis caused by pathogenic *Vibrio* is the main cause of disease in shellfish [28]. More than 20 pathogenic *Vibrio* species have been documented to induce the mass death of shellfish in aquaculture systems [29]. The extracellular metalloprotease (*VtpA*) from *V. tubiashii* is toxic to Pacific oyster (*Crassostrea gigas*) larvae [30]. Pathogenic *V. tubiashii* causes the death of *Mactra chinensis* [31] and various zoonoses [32,33,34,35]. Moreover, environmental stress could affect the immune ability of scallops and the immune factor response [36]. Superoxide dismutase (SOD) and catalase (CAT) are important members of the antioxidant reaction mechanism in organisms which can eliminate and balance the action of ROS and are important indicators to detect the immune defense ability of shellfish. Under environmental stress, scallops produce ROS and other harmful substances. SOD can decompose ROS into large amounts of H_2_O_2_, and the H_2_O_2_ can be disintegrated into H_2_O and O_2_ via glutathione peroxidase and CAT [37], which can eliminate the deleterious effects of reactive oxygen species (ROS) on scallops. There is thus a need to develop strategies to prevent vibriosis and enhance the productivity of scallop cultivation.

Given that seaweeds are resistant to *Vibrio*, they could potentially be used to prevent vibriosis outbreaks [38]. In addition, our previous study [39] showed that *U. pertusa* had the most potent *Vibrio* suppressive activity among eight macroalgal species, followed by *G. lemaneiformis*. Therefore, we determined the optimal environmental parameters for the co-culture of macroalgae (*U. pertusa* and *G. lemaneiformis*) with *C. farreri* through measurements of DO, pH, and *Vibrio* number. This study aims to establish one co-culture system for seaweeds and scallops and to inhibit or reduce vibriosis occurrences. Generally, our findings have important implications for the development of commercially significant seaweed and will be applied to scallop and seaweed aquaculture on a large scale in the future.

## 2. Results

### 2.1. Optimal Conditions for Seaweed–Scallop Co-Culture System

In the mono-culture system of *U. pertusa* (Experiment A, Table 1), the DO in the 30 g and 40 g *U. pertusa* treatments was highest at 36 h (18.78 mg L^−1^ and 17.34 mg L^−1^, respectively) (Figure 1a). The DO and pH were higher in the 30 g *U. pertusa* treatment than in the 40 g *U. pertusa* treatment (*p* < 0.05) (Figure 1b), and the inhibition rates of *Vibrio* in the 30 g and 40 g *U. pertusa* treatments were greater than 99% at 72 h; the *Vibrio* inhibition rate of the 30 g *U. pertusa* treatment was slightly greater than that of the 40 g *U. pertusa* treatment (*p* < 0.05) (Figure 1c). Thus, the 30 g *U. pertusa* treatment was used for co-culture in subsequent experiments.

Three scallops were pre-incubated with 30 g *U. pertusa* to determine the aeration conditions for effective *Vibrio* inhibition in Experiment B (Table 1; Figure 2). In the no-aeration treatment, no *C. farreri* were living at 24 h given that the oxygen provided by the 30 g of *U. pertusa* was insufficient to support their basic needs; measurements of DO, pH, and *Vibrio* number were thus not taken after 24 h. In the continuous aeration treatment, DO ranged from 7.73 mg L^−1^ to 8.17 mg L^−1^, and the pH ranged from 7.43 to 8.76 (Figure 2). Under aeration in the dark, DO ranged from 8.30 mg L^−1^ to 19.44 mg L^−1^, and the pH ranged from 8.45 to 9.94 (Figure 2). DO was significantly higher under aeration in the dark than under continuous aeration (*p* < 0.05). The number of *Vibrio* under aeration in the dark decreased sharply from 9.7 × 10^6^ cfu mL^−1^ to 50 cfu mL^−1^ over the 72-h culture period; *Vibrio* number in the continuous aeration treatment decreased from 8.5 × 10^6^ cfu mL^−1^ to 3910 cfu mL^−1^ (Figure 2c). Therefore, we selected aeration (1.25 L min^−1^) during the dark cycles and the absence of aeration during the light cycles. Then, under aeration in the dark, DO and pH were slightly higher with three *C. farreri* individuals compared to four and five *C. farreri* individuals. Additionally, the inhibitory effect on *Vibrio* was more effective with three *C. farreri* individuals than with four and five individuals (Figure 2). As a result, three *C. farreri* individuals were chosen. Finally, the optimal co-culture parameters were 30 g of *U. pertusa*, three *C. farreri*, 6 L of seawater, and aeration in the dark with 12:12 h (L:D), 23 °C, and 60 μmol photons m^−2^s^−1^.

### 2.2. Changes in DO, pH, and Vibrio Number

In total, 30 g of *U. pertusa* and 30 g of *G. lemaneiformis* were co-cultured with three *C. farreri* individuals in Experiment C (Table 1; Figure 3); the co-culture systems under aeration in the dark comprised the experimental groups (CU and CG, respectively), and the co-culture systems under continuous aeration comprised the control groups (cu and cg, respectively). DO increased with the duration of illumination and reached 19.44 mg L^−1^ and 16.58 mg L^−1^ in the CU and CG groups at 12 h of light (Figure 3a), respectively. The pH ranges were 7.92–9.94 and 6.69–9.27 for CU and CG (Figure 3b), respectively. The *Vibrio* number decreased from 1.94 × 10^7^ cfu mL^−1^ (0 h) to 56 cfu mL^−1^ (72 h) in the CU group, and from 1.99 × 10^7^ cfu mL^−1^ (0 h) to 4 × 10^2^ cfu mL^−1^ (72 h) in the CG group (Figure 3c). At most time points, DO and pH were higher in the CU and CG groups than in the cu and cg groups. The decrease in *Vibrio* number in the first 24 h was more pronounced in the CU and CG groups than in the cu and cg groups. The DO at 12 h was higher in the *U. pertusa* mono-culture (19.87 mg L^−1^) than in the *G. lemaneiformis* mono-culture (17.66 mg L^−1^) system (Figure 3a). The *Vibrio* number showed declines from 1.95 × 10^7^ cfu mL^−1^ (0 h) to 60 cfu mL^−1^ (72 h) in the um group, and from 1.95 × 10^7^ cfu mL^−1^ (0 h) to 3.8 × 10^2^ cfu mL^−1^ (72 h) in the gm group. *Vibrio* inhibition was stronger under *U. pertusa* mono-culture than under *G. lemaneiformis* mono-culture. The number of *Vibrio* decreased as the DO and pH increased in the *C. farreri*–*U. pertusa* and *C. farreri* and *G. lemaneiformis* co-culture system, and the decrease in *Vibrio* number was highest in the optimal co-culture system.

### 2.3. Nutrient Changes

In the control groups (Figure 4), the concentrations of four nutrients significantly increased in the *C. farreri* mono-culture system (cm), and they were 0.1407 mg L^−1^ (NH_4_-N), 0.0260 mg L^−1^ (NO_2_-N), 0.1094 mg L^−1^ (NO_3_-N), and 0.1992 mg L^−1^ (PO_4_-P) at 72 h (Figure 4). The concentrations of these nutrients decreased markedly under the *U. pertusa* and *G. lemaneiformis* mono-culture. In the CU and CG groups, the concentrations of these four nutrients increased slightly in the first 24 h and then decreased; this might stem from the increase in the absorption of these nutrients by seaweed as the culture period extended. In the CU and CG groups (Table 1; Figure 4), the concentrations of NH_4_-N, NO_2_-N, NO_3_-N, and PO_4_-P were maintained at approximately 0.0688 mg L^−1^–0.0751 mg L^−1^, 0.0076 mg L^−1^–0.0111 mg L^−1^, 0.0453 mg L^−1^–0.0532 mg L^−1^, and 0.0100 mg L^−1^–0.0238 mg L^−1^ during 48–72 h, respectively. Nutrient concentrations were lower under aeration in the dark than under continuous aeration, and the seaweeds could absorb more nutrients from the co-culture systems than from the mono-culture systems.

### 2.4. Enzyme Activities of C. farreri

In the *C. farreri* mono-culture system, SOD and CAT activity in the adductor muscles, mantle, and liver pancreas decreased slightly at 72 h, although decreases in the activities of these enzymes were not significant (*p* > 0.05) (Figure 5a,b). Compared to 0 h, SOD and CAT activities significantly increased in the CU and CG groups at 72 h (*p* < 0.05) and were higher in the CU and CG groups than in the cu and cg groups (*p* < 0.05) (Table 1; Figure 5a,b). This suggests that the antioxidant and immune activities of scallops were higher under aeration in the dark than under continuous aeration.

Pyruvate kinase (PK) activities significantly increased in the CU and CG groups at 72 h and were higher in the CU and CG groups than in the cu and cg groups at 72 h (*p* < 0.05) (Figure 5c) (*p* < 0.05). PK activities varied among scallop tissues; PK activity was higher in the adductor muscles (151.69 U mg^−1^–381.29 U mg^−1^) than in the gills and mantle. Lactate dehydrogenase (LDH) activities in the adductor muscles, mantle, and gills decreased over the 72 h culture period (*p* < 0.05) (Figure 5d). In the CU and CG groups, LDH activity decreased from 246.31 U g^−1^ (0 h) to 151.29 U g^−1^ (CU) and 154.68 U g^−1^ (CG) in the adductor muscles, from 224.29 U g^−1^ (0 h) to 130.15 U g^−1^ (CU) and 138.21 U g^−1^ (CG) in the mantle, and from 229.22 U g^−1^ (0 h) to 107.74 U g^−1^ (CU) and 108.14 U g^−1^ (CG) in the gills. After co-culture for 72 h, LDH activities were lower under aeration in the dark than under continuous aeration (*p* < 0.05).

### 2.5. Microbial Diversity

The microbial diversity of the experimental and control groups was analyzed using 2,061,778 pairs of reads (Appendix A). A total of 1,860,353 clean reads were generated after quality control and splicing of double-end reads, with an average of 71,552 effective clean reads per sample (Appendix A). All raw data on microbial diversity have been uploaded to the SRA database at NCBI with accession number PRJNA1194227.

After 3 d of co-culture under aeration in the dark, the number of ASVs of the seawater microorganisms dramatically decreased by 68.9% (CU) and 42.9% (CG) (Appendix A), and the abundance of epiphytic microorganisms decreased. Decreases of 34.1% and 24.4% in the number of ASVs of the seawater microorganisms were observed under continuous aeration in the *C. farreri–U. pertusa* and *C. farreri–G. lemaneiformis* co-culture systems.

In the CU and CG groups, the dominant microbial phyla in the seawater and the epiphyte samples were Proteobacteria, Bacteroideta, and Patescibacteria following co-culture for 3 d (Figure 6a). The relative abundance of Bacteroideta increased after 3 d of co-culture (in both seawater and epiphyte samples), but the relative abundance of Proteobacteria and Campylobacterota decreased; Campylobacterota was not detected in *U. pertusa* epiphyte samples. The relative abundance of Patescibacteria increased in the seawater samples but decreased in the *U. pertusa* and *G. lemaneiformis* epiphyte samples. In the control groups, the relative abundance of Patescibacteria decreased under *U. pertusa* and *G. lemaneiformis* mono-culture but increased under *C. farreri* mono-culture.

In the CU group, the relative abundance of *Polaribacter*, *Glaciecola*, unclassified Rhodobacteraceae, and an unclassified-NS9 group increased, and the relative abundance of *Pseudoalteromonas* and *Sulfitobacter* decreased in seawater and epiphyte samples (Figure 6b). In the CG group, the relative abundance of *Sulfitobacter*, unclassified Rhodobacteraceae, and an unclassified NS9 marine group increased, and the relative abundance of *Neptuniibacter* and *Pseudoalteromonas* decreased.

### 2.6. Alpha Diversity and BugBase Phenotypic Predictions

After 3 d of culture, the ACE, Chao1, Simpson, and Shannon indexes declined in all groups (Table 2), which suggests that species richness and diversity decreased in all groups. Under aeration in the dark, some microorganisms, such as Campylobacterota on *U. pertusa*, were absent at the end of the culture period because of the high DO and pH levels (Figure 6a). In the co-culture systems of *C. farreri–U. pertusa* and *C. farreri–G. lemaneiformis* under aeration in the dark, the alpha diversity indexes of microorganisms in seawater were higher than those of epiphytic microorganisms at 3 d. Under *U. pertusa* and *G. lemaneiformis* mono-cultures, the alpha diversity indexes of seawater microorganisms were lower than those of seaweed epiphytic microorganisms.

The abundance of aerobic microorganisms and anaerobic microorganisms increased and decreased, respectively, in the *U. pertusa* and *G. lemaneiformis* mono-culture groups; the opposite pattern was observed in the *C. farreri* mono-culture group (Figure 7a). Under aeration in the dark, the relative abundance of anaerobic microorganisms decreased, and the relative abundance of aerobic microorganisms increased from 28.99% to 38.46% (CU group) and from 27.65% to 37.01% (CG group). High DO levels likely induced increases in the relative abundance of aerobic microorganisms and decreases in the relative abundance of anaerobic microorganisms. In the CU group, there was a high proportion of potentially pathogenic microorganisms on the surface of *U. pertusa* before culture (CUE0) (Figure 7b). Moreover, the abundance of potentially pathogenic microorganisms decreased from 0.61% to 0.09% in the CU group and from 0.23% to 0.15% in the CG group (Figure 7b).

### 2.7. Vibrio Number and Relative Abundance

In the CU and CG groups (Table 3 and Appendix A), the abundance of *Vibrio* decreased after 3 d of co-culture, and the number of *Vibrio* on the thiosulfate–citrate–bile salts–sucrose (TCBS) also decreased (Table 3). The *Vibrio* inhibition rate was 90.9–95.3%, and this was higher than that of the continuous aeration treatment (34.8–83.7%). This suggests that the increase in DO and pH levels effectively inhibited the growth of *Vibrio*.

## 3. Discussion

Seaweeds have a variety of active substances that inhibit *Vibrio* species, such as polysaccharides, lipids, phenols, and terpenoids, which have strong activity [40,41,42,43]. It has also been reported that living algal bodies can inhibit *Vibrio* [6,9]. However, little attention has been paid to the introduction of seaweeds into aquatic animal culture systems to analyze the prevention and control of vibriosis and the changes in the microbial community. At present, scallop farming is an activity of high value in the coastal countries of the world and has further become one of the main ways to utilize marine resources in these countries and regions [44]. Mariculture shellfish, as an important economic type in aquaculture industry, brings huge benefits to the economic income of various countries [45]. Although various breeding models of *C. farreri* have been extensively investigated, disease outbreaks in *C. farreri*, especially vibriosis, are still not effectively addressed [46,47]. Due to the gradual increase in global temperature, the proliferation of *Vibrio* bacteria directly affects the physiological state of scallops and other ponderable marine organisms, leading to a serious decline in the yield and quality of aquatic products [23]. Therefore, it is necessary to design a reasonable scallop culture model. As a metric for bacterial respiration, DO levels have been shown to be inversely correlated with *Vibrio* levels [13,48]. Oxygen produced by seaweed promoted the inhibition of *Vibrio* in the co-culture systems. Our study used co-culture involving green macroalgae (*U. pertusa*) and red (*G. lemaneiformis*) macroalgae with Chinese scallop (*C. farreri*) to optimize reasonable co-culture systems that effectively inhibit *Vibrio* by increasing DO and pH values. Here, the optimal co-culture system comprised three Chinese scallops (*C. farreri*) and 30 g of seaweed (*U. pertusa*/*G. lemaneiformis*) in 6 L of seawater under aeration in the dark (1.25 L min^−1^, 23 °C, 60 μmol photons m^−2^ s^−1^, 12:12 h L:D, 32 × 24 × 20 cm plastic box), as the inhibition of *Vibrio* was strongest in this treatment, which has positive effects on the growth of scallops. *Vibrio* inhibition was more effective under aeration in the dark than under continuous aeration (both light and dark cycles) for the *C. farreri–U. pertusa* and *C. farreri–G. lemaneiformis* co-culture systems, and aeration in the dark was more energy-efficient compared with continuous aeration. Therefore, the semi-closed co-culture system under aeration in the dark was used for indoor culture that could be effective for scallop culture.

*Vibrio* inhibition was most pronounced when the DO level reached 17.59 mg L^−1^ and the pH was 9.94. Under these optimal co-culture conditions, the inhibition rate of *Vibrio* was as high as 99.99% during 3 days of co-culture. We concluded that the oxygen generated by algal photosynthesis during the light cycles (without aeration) was accumulated to support the respiratory metabolism of scallops, and the increase in the DO level inhibited *Vibrio* in the co-culture system. In the co-culture system with aeration in the dark, the DO and pH levels of *C. farreri–U. pertusa* were slightly higher than those of *C. farreri–G. lemaneiformis* during the incubation process, and *Vibrio* numbers could be reduced by 6 (*C. farreri–U. pertusa*) and 5 (*C. farreri–G. lemaneiformis*) orders of magnitude after 3 days of co-culture. This may be because the leaf surface area of *U. pertusa* was larger than that of *G. lemaneiformis*, which led to a larger light-harvesting area of *U. pertusa* and made the photosynthesis of *U. pertusa* stronger than that of *G. lemaneiformis* and its oxygen production greater. Therefore, *U. pertusa*, with the same biomass, produced more oxygen than *G. lemaneiformis*. The leaf surface area directly modified the exposure of single leaves, leaf arrangement, and aggregation on the shoot, which further significantly impacted the average irradiance on the leaf surface [49,50]. Nevertheless, under continuous aeration, the oxygen produced by algal photosynthesis during the light cycles was driven to the air due to aeration, and the oxygen could not be accumulated, resulting in the failure of DO reaching a concentration that could inhibit *Vibrio*. Furthermore, we first reported the co-culture conditions of aeration in the dark, which not only ensured that the respiration of scallops was unaffected without oxygen supply from seaweed photosynthesis but also allowed oxygen accumulation to improve DO levels during the day, achieving a better *Vibrio* inhibition effect.

The nutrients in seawater mainly include NH_4_-N, NO_3_-N, NO_2_-N, and PO_4_-P [51,52]. As a harmful substance mainly to marine animals, NH_4_-N is produced by the decomposition of organic matter such as aquatic animal excreta, biological carcasses, and bait residues from microorganisms [53]. It can affect the activity of enzymes in aquatic animals, and then affect their development and growth [54]. Widman et al. [55] indicated that ammonia nitrogen was the most toxic nitrogen-containing waste to Juvenile Bay Scallops (*Argopecten irradians*) in the bay. As an intermediate product of nitrification and denitrification, NO_2_-N could inhibit the activity of important metabolic enzymes in organisms when it is enriched to a certain extent, which leads to metabolic dysfunction, physical decline, and large outbreaks of disease and death in shellfish [56]. The high content of phosphorus in seawater will lead to eutrophication of seawater and cause red tides, which will affect the growth of scallops [51]. The low content of phosphorus will affect the growth of phytoplankton in seawater, resulting in a lack of food for scallops in the breeding process, thus affecting breeding benefits [51]. In the optimized co-culture systems of CU and CG, the NH_4_-N, NO_3_-N, NO_2_-N, and PO_4_-P concentrations increased in the first 24 h of the culture period and then decreased gradually below pre-culture concentrations. Nutrient concentrations under aeration in the dark were consistently lower than those under continuous aeration. This might indicate that continuous aeration limits nutrient uptake by algae. The applied *U. pertusa* and *G. lemaneiformis* could absorb NH_4_-N, NO_3_-N, NO_2_-N, and PO_4_-P and reduce eutrophication in seawater in the co-culture system under aeration in the dark.

SOD and CAT are momentous antioxidant enzymes in organisms [57], and their content changes could reflect the oxidative stress level of scallops and indicate the immune defense ability of shellfish at the biochemical level [58]. Under aeration in the dark, the SOD and CAT activities increased over 3 d of co-culture, which indicates that *C. farreri* exhibited positive immunological responses to aeration in the dark. High DO and pH levels induced stress responses in scallops, and the increase in SOD activity promoted ROS scavenging, which generated large amounts of H_2_O_2_. The H_2_O_2_ was then disintegrated into H_2_O and O_2_ via glutathione peroxidase and CAT [37], which eliminated the deleterious effects of ROS on scallops. Over 3 d of co-culture, PK activities in the adductor muscles, mantle, and gills increased, and LDH activities decreased. PK activities were higher in the adductor muscles than in the mantle and gills; this likely stems from the fact that the adductor muscles require more energy than the mantle and gills given that they are responsible for the opening and closing of the shell [59]. In conclusion, variation in the activities of these four enzymes indicates that the immune and metabolic activities of *C. farreri* were enhanced in the optimal co-culture system.

Bacteria are critical ecosystem drivers in both aquatic and terrestrial ecosystems and are highly diverse and complex organisms that underpin biogeochemical cycles across diverse ecosystems [60,61,62]. In the optimal co-culture system, the abundance of *Vibrio* decreased in the seawater and seaweed epiphyte samples during the 3 d co-culture period. Patescibacteria are known to have minimal biosynthetic and metabolic pathways and are able to attach to multiple hosts for just long enough to loot or exchange supplies [63]. We speculate that its simple metabolic pathways and low-energy attachment mechanism facilitate adaptation to the optimal co-culture system; this also explains the increase in the relative abundance of Patescibacteria over the 3 d co-culture period. Similarly, many studies have shown that the relative abundance of ultra-small Patescibacteria lineages frequently exceed those of other bacteria in diversity surveys [63,64,65,66,67]. Proteobacteria was the second largest phylum of hydrogenogenic CO oxidizers [68], and the decrease in Proteobacteria after 3 days allowed the scallops and seaweed to maintain a good physiological state in the optimal co-culture system. Given that some *Bacteroides* are aerotolerant [69], the relative abundance of *Bacteroideta* significantly increased in the co-culture system with high DO levels. Based on the BugBase phenotypic prediction, the relative abundance of aerobic bacteria increased, whereas that of anaerobic bacteria decreased after 3 days of culture. This may be related to the increase in DO levels in the co-culture system. A high proportion of potentially pathogenic microorganisms appeared in CUE0, suggesting that the potential pathogenic microorganisms might come from *U. pertusa*. Most importantly, the potential pathogenicity of the microorganisms attenuated in the optimal co-culture systems after 3 days of culture, which further proved the effectiveness of our designed co-culture systems. Thus, the co-culture of seaweeds and scallops on large scales could inhibit *Vibrio* and promote scallop growth.

## 4. Materials and Methods

### 4.1. Scallops, Seaweeds, and Vibrio Species

Live *C. farreri* was obtained from a fish market in Qingdao, Shandong, China; healthy scallops (shell height: 3–5 cm; wet weight: 32 ± 5 g) were used in the experiments. After washing the scallops with seawater to remove debris, three scallops were placed in each plastic box (32 × 24 × 20 cm, a total of 9 plastic boxes) with 6 L of freshly filtered seawater (through 0.44 μm microporous membranes) aerated at 1.25 L min^−1^ at 23 °C. *Ulva pertusa* (containing holdfasts and fronds with irregular holes) was collected from the intertidal zone in Huiquan Bay (120°20′32.6″ E, 36°3′24.0″ N), Qingdao, China; *G. lemaneiformis* was obtained from a farm in Putian, Fujian, China. All the seaweed samples were washed with fresh seawater and pre-cultured for 3 d in a plastic box (32 × 24 × 20 cm) at 23 °C; 50% fresh seawater was exchanged daily under natural illumination. In the experiments, a total of three *Vibrio* strains maintained in the laboratory were utilized: *V. tubiashii*, *V. splendidus*, and *Vibrio* sp. MM5. *V. tubiashii* was previously isolated from *Argopecten irradians* larvae in 1989, *V. splendidus* was obtained from the liver of diseased Chinese white shrimp (*Penaeus chinensis*) in 1990, and *Vibrio* sp. MM5 was isolated from moribund clam (*Meretrix meretrix*) in 2007.

### 4.2. Determination of Vibrio Quantity, DO, and pH

The protocol for inoculating *Vibrio* suspensions was as follows: *V. tubiashii*, *V. splendidus*, and *Vibrio* sp. MM5 stored in 2216E agar slant culture medium were streaked on TCBS plate medium and cultured at 23 °C for 24 h. Subsequently, the single colonies with smooth edges were selected, re-streaked, and cultured for 24 h at 23 °C. The single colonies with smooth edges were then inoculated into 900 mL of 2216E liquid medium and underwent shaking in an incubator at 150 r min^−1^ (23 °C, 24 h). Next, the *Vibrio* precipitate was collected by centrifugation at 1776× *g* (23 °C, 20 min) and thoroughly resuspended using 10 mL of sterilized seawater (0.44 μm filter membrane, high-temperature autoclaving) to prepare three *Vibrio* suspensions.

To determine the number of *Vibrio*, 1 mL of stirred seawater sample was diluted 1–6 times (m) and 100 μL of diluted seawater was coated on the TCBS selective medium (Solarbio, Beijing, China). The inoculated TCBS plate was then incubated at 23 °C for 48 h, and the number of *Vibrio* colonies (n) was counted. All of the above operations were performed on an ultra-clean workbench. Three cycles of TCBS medium were used for each sample.*Vibrio* count = n × 10^m+1^ cfu mL^−1^(1)

The DO and pH levels of the seawater were measured using a FireSting-O_2_ m (Beijing Ecotech Ecological Technology Ltd., Beijing, China) and METTLER TOLEDO pH meter (FE20, Shanghai Mettler Toledo Ltd., Shanghai, China), respectively. Air in the co-culture was provided by an air pump (Zhongshan SOBO, Electric Appliance Co. Ltd., Zhongshan, China). Data on the number of *Vibrio*, DO level, and pH level were measured every 12 h during the 72 h culture period.

### 4.3. Experimental Design

We conducted a series of experiments to determine the optimal co-culture conditions for inhibiting *Vibrio* (Table 1). Firstly, we aimed to determine the optimal amount of seaweed for *Vibrio* inhibition in Experiment A. In total, 0 g, 30 g, and 40 g of fresh *U. pertusa* were added to each plastic box (32 × 24 × 20 cm) containing 6 L of seawater (without scallops), and 0 g *U. pertusa* served as the control group. The *Vibrio* suspensions (about 10^7^ cfu mL^−1^) were then added into each plastic box. Comprehensively considering the alterations in the three factors (*Vibrio* number, DO value, and pH value) and the effect of mutual shading by seaweed in the plastic boxes, the optimal algae density that produced the most pronounced effect of decreasing the number of cultivable *Vibrio* was determined. Secondly, in Experiment B, we aimed to determine the optimal aeration conditions and scallop number that inhibited *Vibrio* bacteria and promoted scallop growth. The initial concentration of *Vibrio* suspensions was about 10^7^ cfu mL^−1^. We selected three scallops and 30 g *U. pertusa* for the co-culture to explore the optimal aeration conditions. The co-culture system of *C. farreri–U. pertusa* was subjected to three sets of aeration conditions by using air stones to ad air: no aeration, continuous aeration (1.25 L min^−1^) during light and dark cycles, and aeration (1.25 L min^−1^) during the dark cycles and the absence of aeration during the light cycles. Under the optimized ventilation conditions explored above, three, four, and five scallops (*C. farreri*) were used to determine the optimal number of scallops with 6 L of seawater and 30 g or *U. pertusa*. Then, in Experiment C, we evaluated the efficacy of the optimal amount of seaweed and scallops and the aeration conditions identified in Experiments A and B for the co-culture of *C. farreri* with *U. pertusa* and *G. lemaneiformis*. Control and experimental groups (with the addition of *Vibrio* suspensions, about 10^7^ cfu mL^−1^) were used to determine the extent to which these optimal conditions inhibited *Vibrio*. Finally, in Experiment D, we evaluated the effectiveness of applying these optimal conditions to a normal aquaculture environment (without adding *Vibrio* suspensions) by measuring the enzyme activities of the scallops, nutrient salts, the abundance of *Vibrio*, and microbial diversities.

In Experiment A, B, and C, the *Vibrio* number, DO value, and pH value were measured every 12 hrs until the end of the 72 hrs. Details on all experiments (Experiment A, B, C, D) are provided in Table 1. There were three replicates in each group, and we used 12 h:12 h light/dark cycles and 60 μmol photons m^−2^s^−1^ (LED).

### 4.4. Nutrient Detection

In Experiment D, all groups were pre-cultured for 1 h under the same culture conditions. A total of 200 mL of seawater samples was collected at 24-h intervals until the end of the 72-h period, and seawater was stirred to insure homogenization before sampling. These seawater samples were used to determine the ammonia nitrogen (NH_4_-N), nitrate nitrogen (NO_3_-N), nitrite nitrogen (NO_2_-N), and inorganic phosphorus (PO_4_-P) content. Measurements of these nutrients were conducted according to national standards for marine sampling [70]. NH_4_-N was measured using the indophenol blue method, NO_2_-N was measured using *N*-naphthylethylenediamine dihydrochloride, and PO_4_-P was measured using the phosphorus-molybdenum blue method [71,72]. NO_3_-N was measured using the cadmium column reduction method with a reduction column (LS_3105) obtained from Qingdao Anlixin Trading Co., Ltd. (Qingdao, China).

The NH_4_-N, NO_2_-N, NO_3_-N, and PO_4_-P concentrations were measured in Experiment D, and they were calculated using the following formulas (Appendix A):NH_4_-N: y = 0.6829x − 0.0007  r^2^ = 0.9998(2)NO_3_-N: y = 1.795x + 0.009  r^2^ = 0.9970(3)NO_2_-N: y = 2.760x − 0.001  r^2^ = 0.9992(4)PO_4_-P: y = 0.435x + 0.001  r^2^ = 0.9990(5)

### 4.5. Enzyme Activity Assays

In Experiment D, the activities of SOD, CAT, PK, and LDH in the scallops were determined to evaluate the immune and metabolic activities of the scallops. Crude enzyme extracts were obtained from the adductor muscles, mantles, liver pancreas, and gills, and enzyme activities were determined at 0 h and 72 h during the co-culture period. All three scallops from each box were collected into a sterilized 2 mL centrifuge tube as a replicate, and each group was replicated with three plastic boxes. The extracted crude protein was obtained using a BCA kit (Solarbio, Beijing, China) per the manufacturer’s instructions. The SOD and CAT activities of the adductor muscles, liver pancreas, and mantle were measured, and the PK and LDH activities of the adductor muscles, mantle, and gills were measured. The activities of all enzymes were measured using enzyme detection kits (Nanjing Jiancheng Bioengineering Institute, Nanjing, China) per the manufacturer’s instructions.

### 4.6. Microbial Diversity Analysis

In Experiment D, microorganisms in the seawater and on seaweeds (epiphytic microorganisms) were collected before and 3 d after culture, and details on the 26 microbial samples are summarized in Appendix A. The microorganisms from the seawater of each plastic box were filtered through 0.22 μm microporous membranes. Epiphytic microorganisms on the seaweed surface were sampled by adding 30 g of fresh seaweed into a box with 1 L of sterilized seawater (high-temperature autoclaving) with sterilized glass beads (ca. 2 mm in diameter). They were then shaken in an incubator at 1776× *g* (60 min, 23 °C), and the seawater was filtered through 0.22 μm microporous membranes. The samples were then placed in sterilized centrifuge tubes, snap-frozen in liquid nitrogen for 3–5 min, and stored at −80 °C. All 26 samples were plated on TCBS plates before filtration to determine *Vibrio* number, and each sample used three TCBS plates as replicates.

Total genomic DNA was extracted from the 26 samples using the TGuide S96 Magnetic Soil/Stool DNA Kit (TIANGEN Biotech, Beijing, China) per the manufacturer’s instructions, and the DNA concentration and purity were detected using a NanoDrop 2000 UV-Vis spectrophotometer (Thermo Scientific, Wilmington, DE, USA). The V3-V4 hypervariable region of the bacterial 16S rRNA gene was amplified using the primer pair F: 5′-ACTCCTACGGGAGGCAGCA-3′ and R: 5′-GGACTACHVGGGTWTCTAAT-3′; the contents of the PCR are shown in Appendix A. PCR products were purified using the Omega DNA purification kit (Omega Inc., Norcross, GA, USA). The amplicon library was paired-end sequenced (2 × 250) on an Illumina NovaSeq 6000 platform (Beijing Biomarker Technologies Co., Ltd., Beijing, China). Raw data were filtered using Trimmomatic v0.33 [73]; primer sequences were removed using Cutadapt v1.9.1 [74] to obtain the clean reads. The DADA2 method [75] in QIIME2 v 2020.6 [76] was used to denoise the data after quality control to generate amplicon sequence variants (ASVs), and low-abundance ASVs (less than 0.005%) were removed. Taxonomic annotations of ASVs were determined using the Bayesian classifier algorithm and the SILVA database (http://www.arb-silva.de (accessed on 21 July 2023)) [77]. QIIME was used to determine the abundance of each species in samples and generate the species distribution histogram based on the composition of ASVs.

Alpha diversity was analyzed using QIIME2, and species diversity, the Ace, Chao1, Shannon, and Simpson indices were calculated in R software 4.1.0. BugBase [78] was used to predict the organism-level coverage of functional pathways within complex microbiomes as well as biologically interpretable phenotypes. BugBase first normalized operational taxonomic units through prediction of the 16S copy number, and microbial phenotypes were predicted using precalculated files with a threshold of 0.01.

### 4.7. Statistical Analysis

The statistical analysis was conducted utilizing a completely randomized design with three replicates. For the statistical analysis, SPSS version 21 was utilized. To compare the means, one-way ANOVA and Tukey’s HSD’s multiple range tests were employed with a significance level of *p* < 0.05.

## 5. Conclusions

For the first time, optimal co-culture systems that inhibit *Vibrio* and promote the growth of scallops were established using *U. pertusa* and *G. lemaneiformis* with *C. farreri*. Specifically, DO levels, pH levels, and the number of *Vibrio* were measured in this experiment that aimed to determine the optimal amount of seaweed and scallops, the optimal aeration conditions, and the optimal co-culture system for inhibiting *Vibrio*. Finally, we found that a co-culture system comprising three scallops with 30 g of fresh seaweed in 6 L of seawater with aeration (1.25 L min^−1^) in the dark was optimal for decreasing *Vibrio* abundance after 3 days of co-culture. The effectiveness of this optimal co-culture system was verified from the perspectives of nutrient level, enzyme activity, and microbial diversity. The microbial communities in seawater and on algae were altered probably due to the high DO and pH levels of the co-culture system. Based on the BugBase phenotypic prediction, a decrease in pathogenic microorganisms was observed after 3 days of co-culture. Overall, this optimal co-culture system could significantly inhibit *Vibrio*, which enhanced the immune defense and metabolism of scallops. Although previous studies have shown that seaweeds are resistant to *Vibrio*, it remains unclear whether algae co-cultured with scallops can effectively inhibit *Vibrio*. Our study fills this research gap and provides a theoretical basis for vibriosis prevention in the co-culture system of scallop–seaweed, thereby aiding the development of seaweeds with commercial potential. This study may guide the development of comprehensive cultivation models for algae and other aquatic animals, providing a system to effectively restrain *Vibrio* and prevent outbreaks of vibriosis in aquatic animals and promoting the healthy and sustainable development of aquaculture.

## Figures and Tables

**Figure 1 plants-14-00334-f001:**
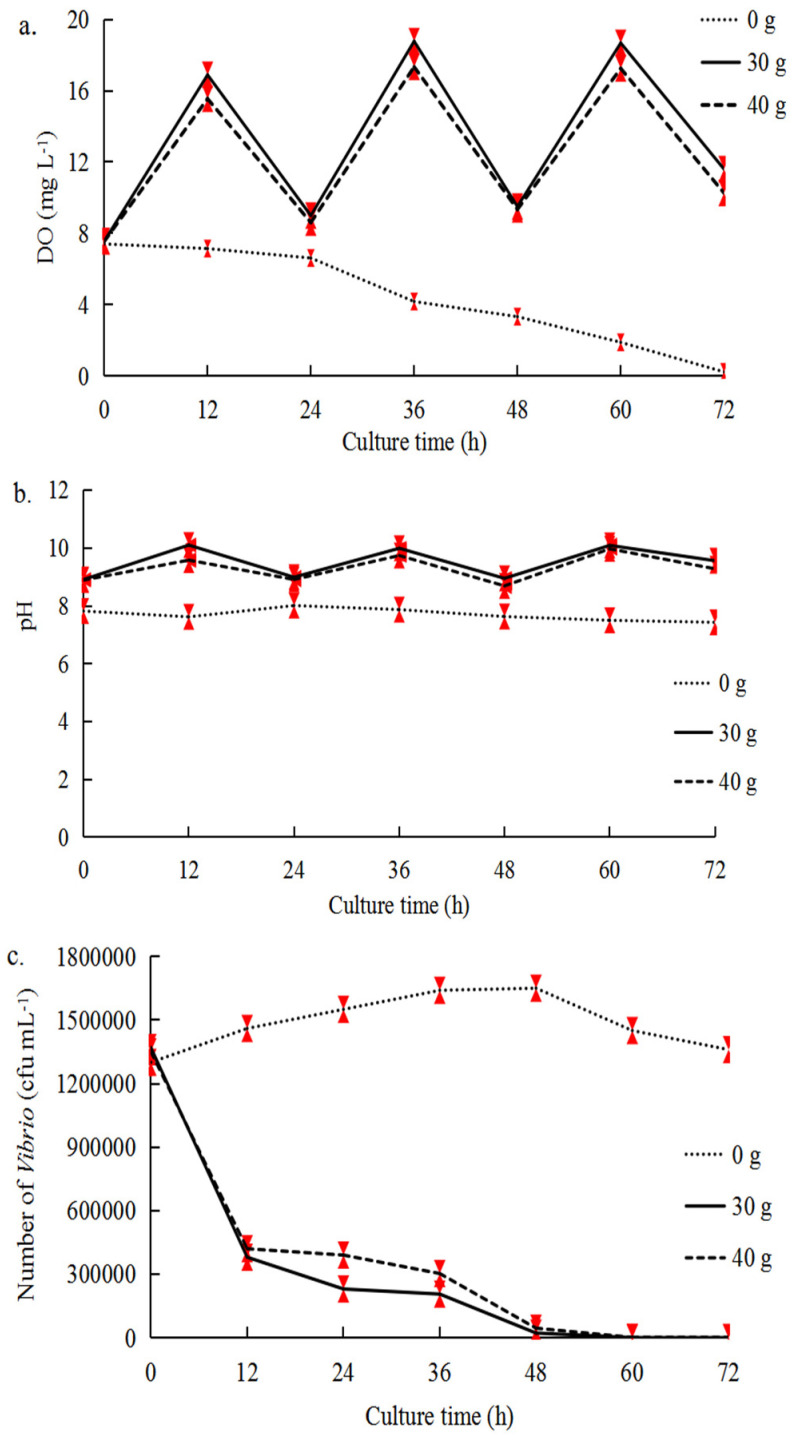
Dissolved oxygen (DO) (**a**), pH (**b**), and *Vibrio* number (**c**) with 0 g, 30 g, and 40 g *U. pertusa* in Experiment A with the addition of three *Vibrio* suspensions (10^7^ cfu mL^−1^). The error bars (red) stand for the standard deviation, n  =  3 (*p* < 0.05).

**Figure 2 plants-14-00334-f002:**
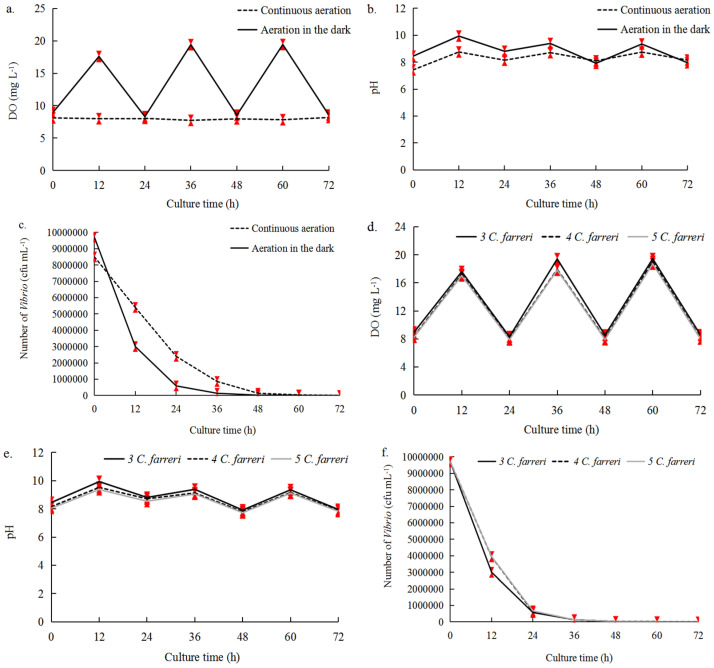
DO, pH, and *Vibrio* number in Experiment B with the addition of three *Vibrio* suspensions (10^7^ cfu mL^−1^). (**a**–**c**) In the co-culture system of 30 g *U. pertusa*–3 *C. farreri* with different aeration conditions; (**d**–**f**) 30 g of *U. pertusa* was co-cultured with 3, 4, and 5 *C. farreri* under aeration in the dark. The error bars (red) stand for the standard deviation, n  =  3 (*p* < 0.05).

**Figure 3 plants-14-00334-f003:**
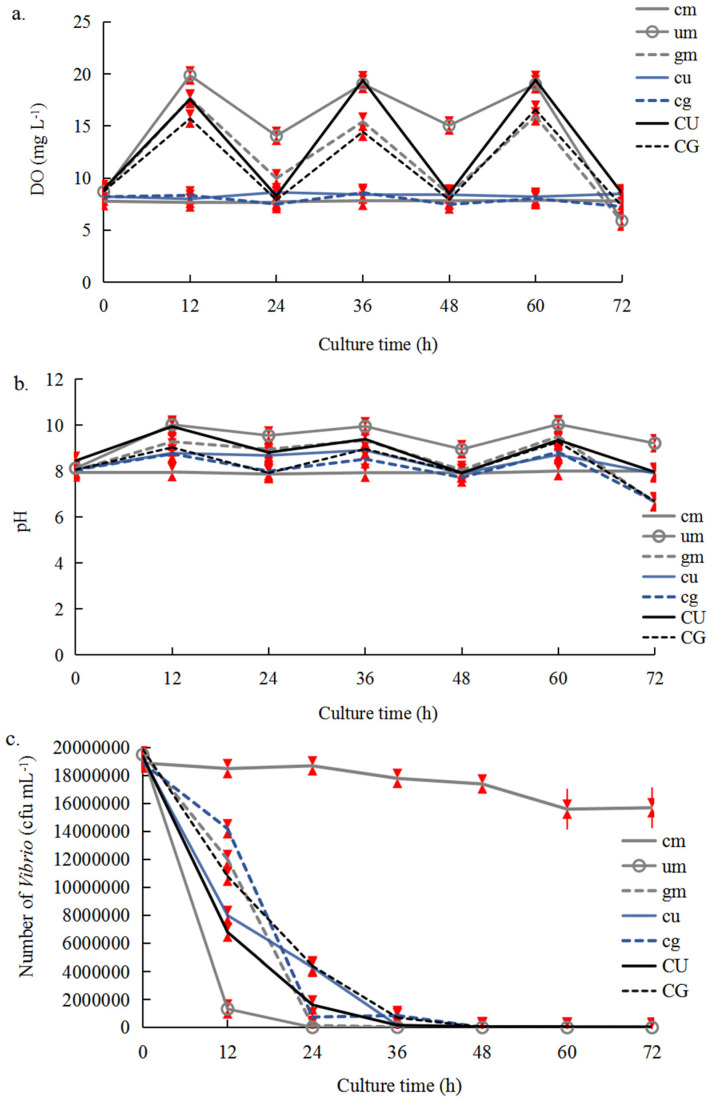
DO (**a**), pH (**b**), and *Vibrio* number (**c**) in the seawater of Experiment C with the addition of three *Vibrio* suspensions (10^7^ cfu mL^−1^). The error bars (red) stand for the standard deviation, n  =  3 (*p* < 0.05). cm: mono-culture of *C. farreri* with continuous aeration; um: mono-culture of *U. pertusa* without aeration; gm: mono-culture of *G. lemaneiformis* without aeration; cu: co-culture of *C. farreri–U. pertusa* under continuous aeration; cg: co-culture of *C. farreri–G. lemaneiformis* under continuous aeration; CU: co-culture of *C. farreri–U. pertusa* under aeration in the dark; CG: co-culture of *C. farreri–G. lemaneiformis* under aeration in the dark. CU and CG were the experimental groups; the others were the control groups.

**Figure 4 plants-14-00334-f004:**
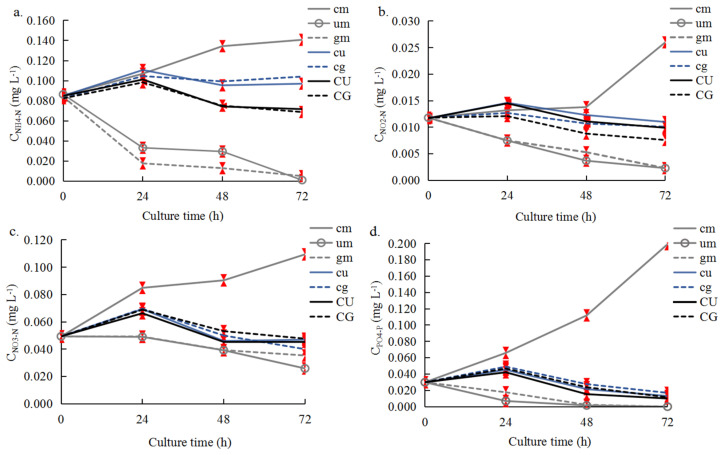
The concentrations of NH_4_-N (**a**), NO_2_-N (**b**), NO_3_-N (**c**), and PO_4_-P (**d**) in Experiment D. Symbols and culture conditions are as described in Figure 3. The error bars (red) stand for the standard deviation, n = 3 (*p* < 0.05).

**Figure 5 plants-14-00334-f005:**
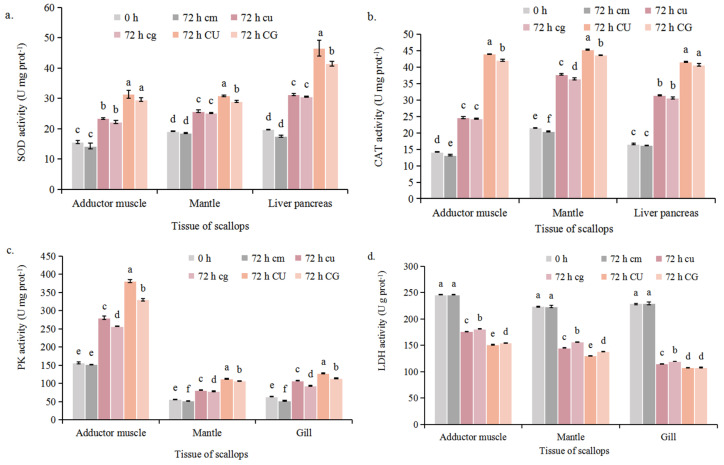
The SOD (**a**), CAT (**b**), PK (**c**), and LDH (**d**) activity of *C. farreri* in Experiment D. Mean ± SD, n = 3. Different letters show the significant difference with *p* < 0.05 using one-way ANOVA and Tukey’s HSD tests. The symbols and culture conditions are as described in Figure 3.

**Figure 6 plants-14-00334-f006:**
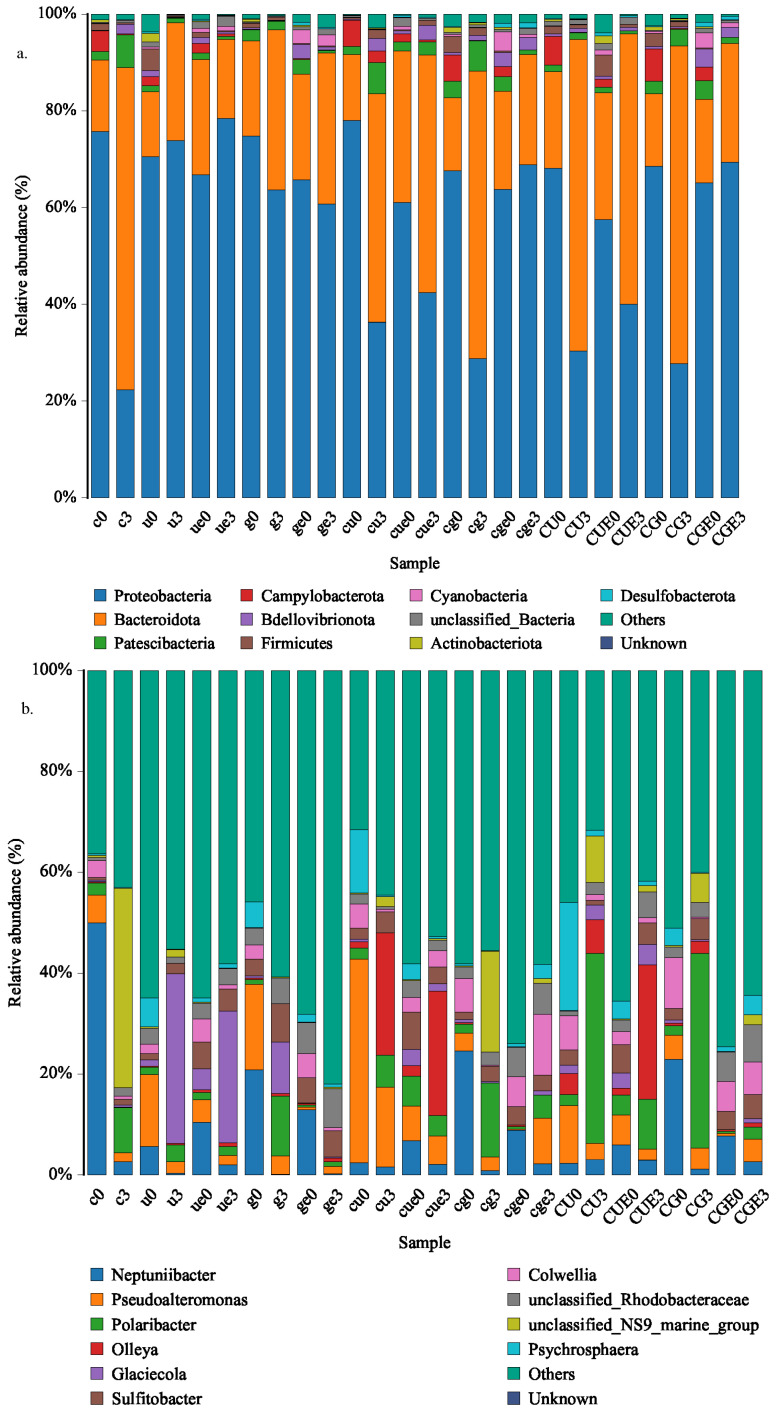
A histogram of the 10 species with the highest relative abundance at the phylum (**a**) and genus (**b**) levels of 26 samples from the seawater environment and seaweed epiphytes. 0: before culture; 3: after 3 days of culture; e: epiphytic microorganisms on seaweeds; without “e”: microorganisms in seawater; c: the mono-culture of *C. farreri*; u: the mono-culture of *U. pertusa*; g: the mono-culture of *G. lemaneiformis*; cu and cg: the co-culture of *C. farreri–U. pertusa* and *C. farreri–G. lemaneiformis* under continuous aeration, respectively; CU and CG: the co-culture of *C. farreri–U. pertusa* and *C. farreri–G. lemaneiformis* under aeration in the dark, respectively.

**Figure 7 plants-14-00334-f007:**
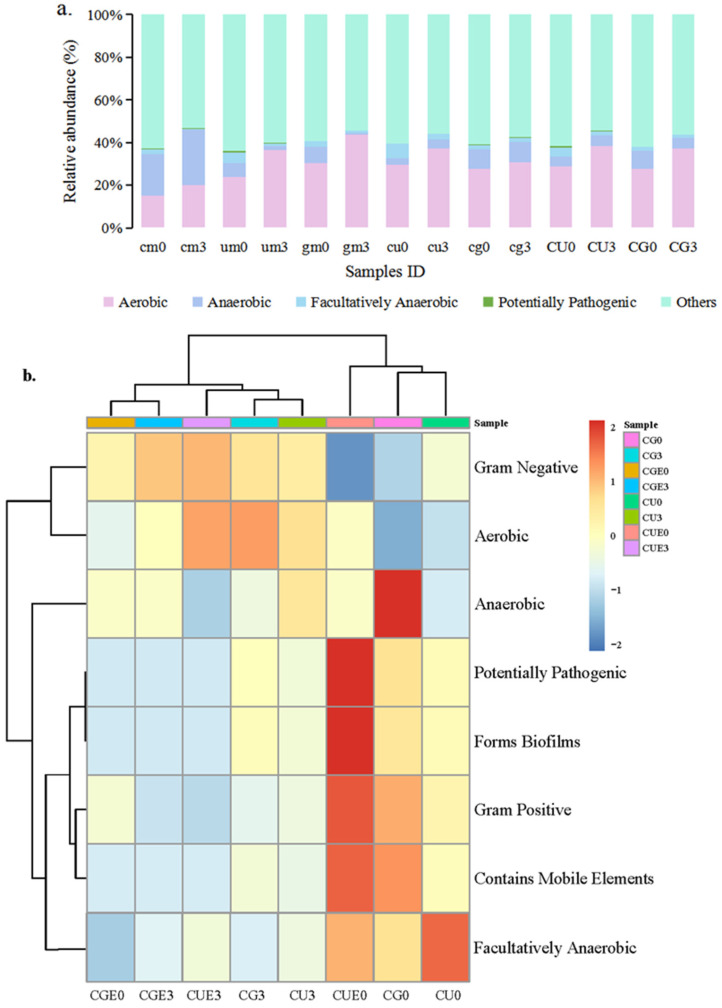
The histogram (**a**) and heat map (**b**) of the phenotype prediction results. (**a**) Seawater and epiphytic microorganisms from each group were pooled for analysis. (**b**) Seawater and epiphytic microorganisms for the CU and CG groups. The symbols and culture conditions are as described in Figure 6.

**Table 1 plants-14-00334-t001:** Specific treatments for Experiments A, B, C, and D.

Treatment ^a^		Algal Weight (g)	*C. farreri* Numbers	*Vibrio* Suspensions (10^7^)	Aeration Condition (1.25 L min^−1^)	Symbol
Expt A		0 g *U. pertusa*	0	Yes	No aeration	-
		30 g *U. pertusa*	0	Yes	No aeration	-
		40 g *U. pertusa*	0	Yes	No aeration	-
Expt B	Aeration condition	30 g *U. pertusa*	3	Yes	No aeration	-
		30 g *U. pertusa*	3	Yes	Aeration in the dark	-
		30 g *U. pertusa*	3	Yes	Continuous aeration	-
	*C. farreri* numbers	30 g *U. pertusa*	3	Yes	Aeration in the dark	-
		30 g *U. pertusa*	4	Yes	Aeration in the dark	-
		30 g *U. pertusa*	5	Yes	Aeration in the dark	-
Expt C	Control groups	0 g	3	Yes	Continuous aeration	cm
		30 g *U. pertusa*	0	Yes	No aeration	um
		30 g *G. lemaneiformis*	0	Yes	No aeration	gm
		30 g *U. pertusa*	3	Yes	Continuous aeration	cu
		30 g *G. lemaneiformis*	3	Yes	Continuous aeration	cg
	Experimental groups	30 g *U. pertusa*	3	Yes	Aeration in the dark	CU
		30 g *G. lemaneiformis*	3	Yes	Aeration in the dark	CG
Expt D	Control groups	0 g	3	No	Continuous aeration	cm
		30 g *U. pertusa*	0	No	No aeration	um
		30 g *G. lemaneiformis*	0	No	No aeration	gm
		30 g *U. pertusa*	3	No	Continuous aeration	cu
		30 g *G. lemaneiformis*	3	No	Continuous aeration	cg
	Experimental groups	30 g *U. pertusa*	3	No	Aeration in the dark	CU
		30 g *G. lemaneiformis*	3	No	Aeration in the dark	CG

^a^ The above experimental treatments groups were incubated in 6 L of seawater at 23 °C, with 12 h:12 h light/dark cycles at 60 μmol photons m^−2^s^−1^. There were three replicates for each experimental group.

**Table 2 plants-14-00334-t002:** Alpha diversity index statistics of microorganisms from the seawater and algal epiphytes.

Samples ID ^a^	ACE	Chao 1	Shannon	Coverage
c0	721.20	721.00	5.53	1.00
c3	290.63	290.04	5.19	1.00
u0	994.00	994.00	7.62	1.00
u3	320.00	320.00	5.12	1.00
ue0	836.88	836.04	7.71	1.00
ue3	484.33	484.00	6.33	1.00
g0	687.75	687.05	6.21	1.00
g3	367.45	367.04	5.91	1.00
ge0	632.65	631.15	7.31	1.00
ge3	546.26	545.13	7.08	1.00
cu0	552.61	552.02	5.87	1.00
cu3	364.57	364.03	5.62	1.00
cue0	572.31	572.00	7.43	1.00
cue3	367.00	367.00	6.42	1.00
cg0	783.00	783.00	7.26	1.00
cg3	592.00	592.00	5.98	1.00
cge0	527.34	526.14	7.35	1.00
cge3	538.88	538.08	7.28	1.00
CU0	705.24	705.00	6.78	1.00
CU3	473.00	473.00	5.47	1.00
CUE0	1235.51	1235.07	8.36	1.00
CUE3	384.88	384.11	6.27	1.00
CG0	836.00	836.00	7.31	1.00
CG3	477.00	477.00	5.54	1.00
CGE0	499.86	498.28	7.40	1.00
CGE3	457.62	457.04	7.33	1.00

^a^ Sample IDs are as described in Figure 6.

**Table 3 plants-14-00334-t003:** The relative abundance and absolute numbers of *Vibrio* from the seawater and algal epiphytes.

Sample ID ^a^	Relative Abundance of *Vibrio* (%)	Number of *Vibrio* (cfu mL^−1^)
c0	0.0096	1025
c3	0.0120	8500
u0	0.0143	750
u3	0.0023	0
ue0	0.0086	5800
ue3	0.0046	<10
g0	0.0012	250
g3	0.0001	0
ge0	0.0051	850
ge3	0.0015	<10
cu0	0.0098	6100
cu3	0.0051	3900
cue0	0.0112	8850
cue3	0.0014	1445
cg0	0.0074	3500
cg3	0.0007	1150
cge0	0.0046	1100
cge3	0.0057	1750
CU0	0.0205	2650
CU3	0.0066	125
CUE0	0.0119	1650
CUE3	0.0024	150
CG0	0.0117	2300
CG3	0.0011	130
CGE0	0.0047	1700
CGE3	0.0041	140

^a^ Sample IDs are as described in Appendix A.

## Data Availability

The data will be made available on request. The data are not publicly available due privacy.

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
