# Peer review of "Co-Culturing Seaweed with Scallops Can Inhibit the Occurrence of Vibrio by Increasing Dissolved Oxygen and pH"

_plants, 2025, doi:10.3390/plants14030334_

Round 1
Reviewer 1 Report
Comments and Suggestions for Authors
The authors investigate the effect of co-culturing scallops with seaweeds with regard to reduction of disease-causing Vibrio. While the study is an interesting one, the title is very mis-leading. The authors found the optimal condition to co-culture scallops with seaweeds to reduce Vibrio. The title reads as seaweeds themselves did the job of reducing Vibrio. The conclusion clearly says microbial communities in seawater and on algae were altered by the high DO and pH of the co-culture system, and a decrease in the pathogenic of microorganisms was observed after 3 days of co-culture. So, it’s the pH and the DO which is responsible for the reduction of Vibrio. The entire narrative of the work needs to be changed.
The work cannot be published at its current form. I am recommending rejection and a possibility to re-submit followed by a major revision because of the major comments below:
Other comments: The introduction is poorly organized, not well focused and the story does not flow well. I suggest the authors to follow a storyline to engage the readers. I hope this will improve the readability of their paper and an enjoyable reading experience.
The authors mention that seaweeds are resistant to Vibrio. This generic statement is not true. While Ulva can be well defended against Vibrio, there is evidence that certain other seaweeds can be hot-spot of Vibrios. So, I suggest the authors to include those studies as well. And then justify their choice behind choosing Ulva for this study.
Comments on the Quality of English Language
Needs a lot of improvement. While I could understand the work, the readability needs improvement.
Reviewer 2 Report
Comments and Suggestions for Authors
General
The manuscript describes interesting results on the co-cultivation of seaweeds with scallops as a strategy to inhibit the occurrence of Vibrio species. The experiments are correctly designed to demonstrate the effect of the seaweeds on the reduction of Vibrio spp. in water and to select the most appropriated algae species and the most favourable conditions. Moreover, the effect of parameter of DO and pH were correlated with Vibrio decrease and the effect of co-cultivation on nutrients removal and scallop immune defence was also investigated. Interestingly, the more suitable conditions (e.g. aeration in the dark) were established to enhance Vibrio reduction.
Experiments were conducted in a short period (3 days) that allowed the observation of the effect, which is very promising. However, further experiments should be conducted to monitor the co-cultivation (e.g. algae and scallop growth, nutrients consumption, interactions with microalgae) in a much longer period, before considering an aquaculture application.
Specific
Introduction
Lines 46-70. Writing could be improved, trying to avoid a long succession of short sentences and references.
I miss the background of Chinese scallop cultivation systems (open flow tanks? bottom culture? rafts with netcages?) and how seaweed could be implemented in those systems.
Introduce the enzymatic activities analysed and their relationship with immunity and scallop metabolism. Also introduce the microbial analysis with reference to previous knowledge on scallop or seaweed microbiome and antibacterial activity in seaweeds. Explain why Ulva pertusa and Gracilariopsis lemaneiformis were chosen and considered good candidates for co-cultivation with scallops.
Results
Include in the text the P values to support significant differences between parameters (e.g. DO, pH) for the different treatments (e.g. between 40 and 30 g). Errors bars are missing on the figures. Symbols could be bigger for more clarity.
Line 140. Formulas used for nutrient calculations should go in Materials and Methods, indicating the wavelength used for the measurements of absorbances.
Figure 7. It is surprising to find in CUE0 a high proportion of potentially pathogenic bacteria, which would mean that come from U. pertusa. This fact should be discussed, and the potentially pathogenic functions predicted specified.
Sequences should be deposited in a public repository and the reference or accession number, included.
The lack of replicates in the microbial diversity analysis does not allow any statistical comparison of the treatments (e.g. PERMANOVA) on bacterial community composition or predicted function, and unfortunately this hampers the possibility to elucidate the effect of introducing seaweed on the water of the culture of scallop (e.g. comparing water community in C. farrei monoculture versus co-culture) and only a descriptive approach of the samples analyzed can be done.
Discussion
Line 323. I do not clearly understand the meaning of the phrase: Enzymes were the fundamental driving force….
Microbial diversity analysis should be discussed. Also, the predicted functional results (aerobic/anaerobic/pathogenic,
Line 345. The Patescibacteria had minimal biosynthetic and metabolic pathways and was able to attach multiple hosts…. Do you mean: Patescibacteria are known to have… and to be able…
The presence of other phyla (e.g. Proteobacteria) or genus (e.g. Polaribacter), with are predominant should be also discussed.
Materials and Methods.
Lines 366-368. Describe how were the algae species identified. Many Ulva species are cryptic and should be genotyped for a correct identification.
Were the algae in fragments? maintained in movement?
Line 370. Mention the origin of the Vibrio strains used. I understand that the suspension was a mixture of the three strains. Explain why this was done like that and not one single strain independently.
Line 458. All 26 samples were placed on TCBS… Do you mean plated?
Conclusions
The feasibility is mentioned, here and in other parts of the manuscript, related with nutrient level, enzyme activity and microbial diversity, and for me it is not clear what do you mean. I wonder if you mean the effectiveness.
Line 498. The microbial communities in seawater and on algae were altered by the high
DO and pH of the co-culture system… The two facts are correlated but causality cannot be claimed without further investigation. I would say “probably due to”
Line 499. … and a decrease in the pathogenic of microorganisms was observed after 3 days of co-culture. …. This is based only on predicted functions and should be clearly stated. The type pathogenic bacteria or bacterial functions identified, should be described. BugBase analysis results could be included in Supplementary Materials.
Line 505. Define “integrated culture micro-systems”
Authors should avoid repetition of the results obtained but mention what can be concluded that may guide future work to further elucidate underpinning mechanisms or to develop application protocols in aquaculture.
Round 2
Reviewer 2 Report
Comments and Suggestions for Authors
Lines 43- 47. It still mis some mention to relevant literature related to seaweed microbial interactions in general (e.g. Egan et al 2013), antimicrobial activity in seaweeds (e.g. Vatsos et al 2015) or from seaweeds-associated bacteria (e.g. Singh et al 2015). In particular, those related to Ulva (e.g. Ismail et al. 2018) or to Gracilariopsis (Xie et al. 2017).
Line 82. There is thus a need to develop strategies to prevent vibriosis and enhance the productivity of scallop cultivation. I will move this sentence up, to line 74
Line 85. There is no mention to prevention of vibiosis in the referred paper [38] and I mis some references related to vibrio, or vibriosis prevention by seaweeds (e.g. Pintado et al 2023) including IMTA with mollusks (de Jager et al 2024).
Line 86. In addition, our previous study (unpublished) [39]….
Line 91. Generally, Our findings may have important implications for the development of commercially significant seaweeds and could be applied to scallop and seaweed aquaculture on a large scale in the future.
Line 257 (Results). High proportion of potentially pathogenic microorganisms on the surface of U. pertusa… and Line 380-381 (Discussion). This is a very relevant issue which is not sufficiently clarified. It would hamper the application of co-culture, due to the risk of introducing pathogenic bacteria. The analysis was conducted based on predicted phenotypes using BugBase, presented in Supplementary Material 2. I think those potentially pathogenic phenotypes should be described and discussed in the main text.
Revise tenses:
e.g. Line 366. The Patescibacteria are known to have….Line 372. Proteobacteria are the second largest phylum of hydrogenogenic CO oxidizers [69]